# A construction method of reverberation suppression filter using an end-to-end network

**Zhen Wang**[1], **Hao Zhang**[1,2]*, **Xiao Chen**[1], **Yuan An**[1]

**1** College of Information Science and Engineering, Ocean University of China, Qingdao, Shandong Province, China, **2** Open Studio for Marine High Frequency Communications, Pilot National Laboratory for Marine Science and Technology, Qingdao, Shandong Province, China

\* zhanghao@ouc.edu.cn

**Data Availability Statement:** All code files are available from the figshare database.(https://doi.org/10.6084/m9.figshare.24151551). All relevant data are within the manuscript and its Supporting information files.

## Abstract

Reverberation is the primary background interference of active sonar systems in shallow water environments, affecting target position detection accuracy. Reverberation suppression is a signal processing technique used to improve the clarity and accuracy of echo by eliminating the echoes, reverberations, and noise that occur during underwater propagation.This paper proposes an end-to-end network structure called the Reverberation Suppression Network (RS-U-Net) to suppress the reverberation of underwater echo signals. The proposed method effectively improves the signal-to-reverberation ratio (SRR) of the echo signal, outperforming existing methods in the literature. The RS-U-Net architecture uses sonar echo signal data as input, and a one-dimensional convolutional network (1D-CNN) is used in the network to train and extract signal features to learn the main features. The algorithm's effectiveness is verified by the pool experiment echo data, which shows that the filter can improve the detection of echo signals by about 10 dB. The weights of reverberation suppression tasks are initialized with an auto-encoder, which effectively uses the training time and improves performance. By comparing with the experimental pool data, it is found that the proposed method can improve the reverberation suppression by about 2 dB compared with other excellent methods.

## Introduction

In active sonar detection of underwater targets, the targets are often in a proud or buried state, which means that the target echo will be accompanied by solid reverberation interference at the receiving end of the sonar, which seriously affects the detection of underwater targets [1]. Reverberation is one of the most critical factors affecting sonar detection performance, especially in shallow seas. Reverberation is a random signal generated by the random scattering and superposition of sound waves after encountering many uneven bodies in the undulating sea surface, seabed, and seawater during propagation. The time-frequency reverberation characteristics are related to the transmitting signal, which overlaps with the target echo in the time domain and is coherent with that in the frequency domain [2]. Removing reverberation from

**Funding:** Hao Zhang was supported in part by the National Natural Science Foundation of China under Grant 91938204ï41527901ï61701462. URL of the funder website is 'https://www.nsfc.gov.cn/'. NO, The funders had no role in study design, data collection and analysis, decision to publish, or preparation of the manuscript.

**Competing interests:** The authors have declared that no competing interests exist.

the time or frequency domain independently is difficult. Therefore, effectively reducing the interference of reverberation to target detection has always been a hot topic in underwater acoustic signal processing.

The superimposition of active sonar forms ocean reverberation emission signals after being scattered by many random scatterers and shattering ocean boundaries. However, due to the rapid development of active sonar systems in recent years, the transmitted frequencies are lower than before, and it has been possible to reduce the effects of reverberation effectively. However, the transmission power is higher, so it can detect longer distances. Reverberation is still the most important influencing factor inside the active sonar. Currently, there are many studies on reverberation suppression for transmit signal adjustment. J.P. Costas proposed a frequency hopping signal based on special frequency coding, with ideal Doppler time resolution and good reverberation suppression effect [3]; Henry Cox et al. Collins et al. comprehensively analyzed the anti-reverberation characteristics of various Doppler-sensitive active sonar signals using the Q function [4]; Ward S et al. detected low-Doppler target performance for SFM signals. A detailed analysis was carried out [5].

The ocean reverberation and target echo signals are strongly correlated in the time domain. The two frequency spectra overlap in the frequency domain, and ordinary matched filtering methods cannot effectively find the target echo signal. To improve the performance of coherent processing in reverberation, the reverberation statistical model can suppress the reverberation, whiten the non-Gaussian colored noise into Gaussian white noise, and then detect the signal. Among them, the AR pre-whitening processing method is used to filter the reverberation into white noise under certain conditions to obtain a higher gain through matched filtering and detect the target echo more effectively [6]; It realizes the detection of the time-frequency expansion target, and The target detection performance is improved [7]; the principal component inversion algorithm is used to project the received signal into two subspaces according to the power difference between different backgrounds, thereby realizing the separation of reverberation [8]; based on AR pre-whitening, the received signal is Performed bisection singular value decomposition to achieve a better effect of suppressing reverberation, using the low-rank matrix decomposition method, the received signal is divided into two matrices, a low-rank matrix, and a sparse matrix, to achieve reverberation separation [9].

In recent years, the in-depth development of artificial intelligence technology has brought people hope of solving the reverberation problem. With the continuous development and innovation of deep learning technology, a large number of neural network architectures with good performance and robust stability have emerged, such as Generative Adversarial Networks (GAN) [10], Convolutional Neural Networks (CNN) [11, 12], Recurrent Neural networks (RNN) [13] and so on. These networks often have good performance in different fields. They can solve problems that traditional methods could not solve in the past, attracting scholars from various areas to devote themselves to deep learning research and combining deep learning with their respective fields to provide new ideas for solving problems in their respective fields [14–16].

With the development of artificial intelligence (AI) technology, deep neural networks have brought new research ideas to solve the shallow sea sonar reverberation problem.As a hot research direction in the field of machine learning, the kernel function of the support vector machine is used to detect the signal in the background of reverberation, this method improves the recognition quality of the reverberation background, and its effect is better than the adaptive filtering algorithm [17]; Wu Ketong et al., used a support vector machine to estimate AR, and the coefficient of the model realizes the function of accurately detecting the target signal under the condition of low signal mixing ratio and low Doppler Pan Chengsheng et al. [18]. GAN has become a popular model in the field of deep learning due to its advantages of

generating high-quality samples, learning unlabeled data, supporting multi-modal data and innovation [19, 20]. In the field of underwater acoustic engineering, it is theoretically feasible to use deep learning for active sonar reverberation suppression to solve the problem of reverberation suppression.

In this paper, a neural network, RS-U-Net, which can convolve signals, is constructed using the end-to-end method. The end-to-end neural network is a deep learning model that can directly learn the output from the input data. The end-to-end neural network is trained from input to output. It does not need to extract intermediate features manually and relies on a large enough data set and excellent model design to obtain better results than traditional methods. An end-to-end structure where data is no longer dependent on labels. Let the network discover the components, parse the elements, and restore and amplify the features. The proposed RS-U-Net is a generated encoder-decoder model. The encoder analyzes and learns the reverberation and target echo features during reverberation processing. The decoder part restores the learned features to the signal and generates a reverb suppression model. The target signal initially hidden in the reverb is revealed.

## Related work

This section introduces the implementation of RS-U-Net, and its parts are decomposed and explained here. The general structure of the second section is shown in Fig 1.

Fig 1 presents the framework of the reverberation suppression discussed in the article. The training dataset consists of both pool experiment data and simulation data. The generation of reverberation signals, echo signals, and ocean noise signals is described in the Signal Generation section. Signal processing techniques are explained in the Signal Processing section (similarly applied to pool experiment data). The construction of the end-to-end network structure for generating a reverberation suppression model is explained in the RS-U-Network Construction section. The diagram utilizes two lines to connect each component. The black line represents the training data, while the red line represents the ideal representation of the suppressed

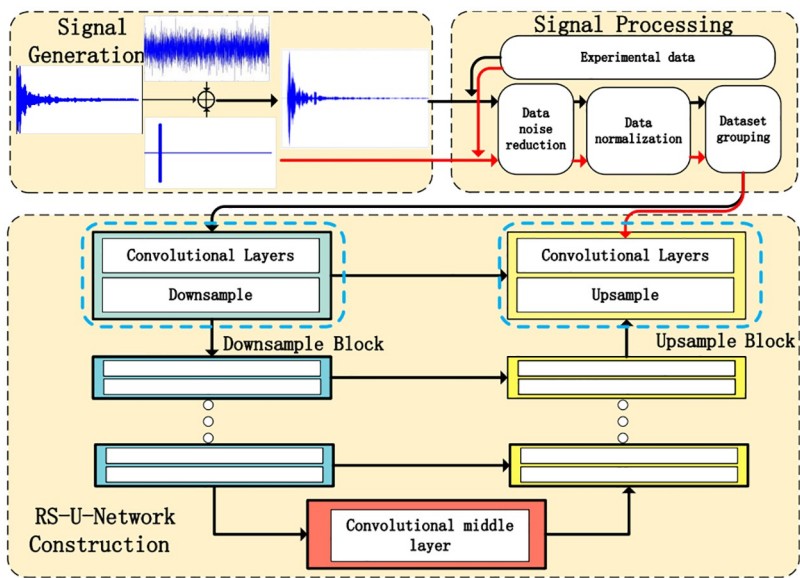

**Fig 1. The overview of the reverberation supperssion framework.**

signal, serving as the validation set. The parameters of the training set can be fine-tuned based on the verification set to implement reverberation suppression models. Finally, the adjusted parameter model is saved and tested using experimental data from the pool.

## Underwater active sonar reverberation simulation

Underwater reverberation can be divided into three components: volume reverberation, sea surface reverberation, and bottom reverberation. Sea surface reverberation and seabed reverberation are collectively referred to as interface reverberation.

A large amount of experimental data is required for network modeling, so sonar reverberation data will be simulated in this paper. Reverberation simulation is divided into three parts: reverberation simulation, echo signal simulation, and environmental noise simulation.

The propagation of the reverberation signal is shown in Fig 2 below.

In Fig 2, Point M sends a non-directional signal for the ring energizer. The figure shows the creation of the reverberation model. *M* is the transducer, the distance from *M* to the interface *xoy* is *h*, and the transducer transmits signals at point *M* without directionality. The signal arrives at the receiver as a spherical extension after reflection from the interface scatterer, at the time *t* inside the ring of scatterers contributing to the reverberation. Over time, the ring gradually moves outward, and its area increases, and therefore the number of scatterers leading to the reverberation increases [21].

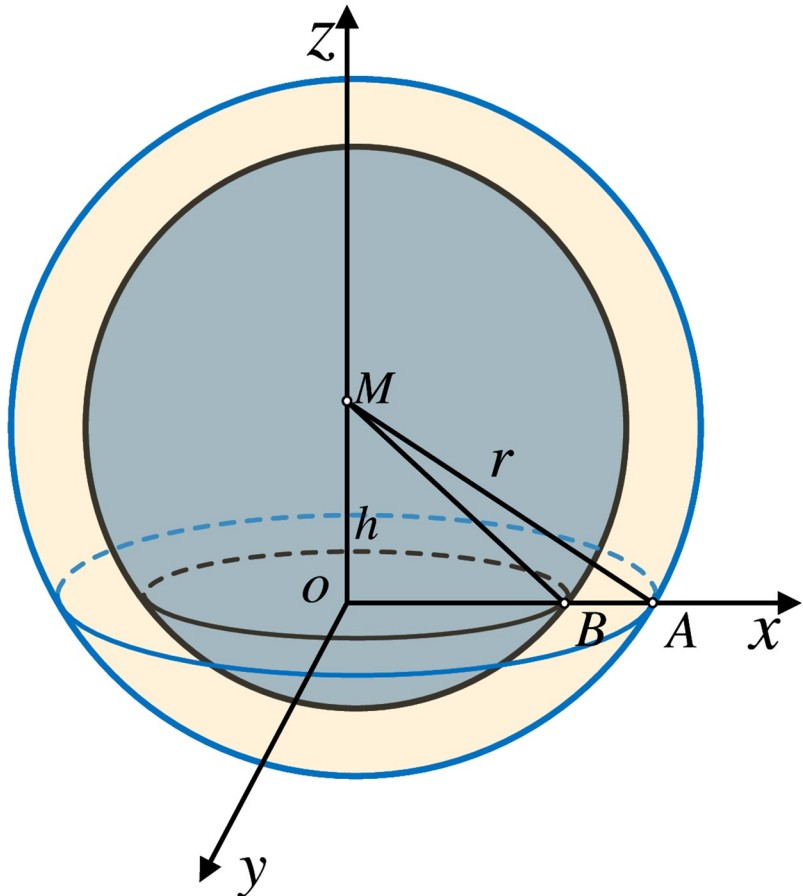

**Fig 2. Interface reverberation model.**

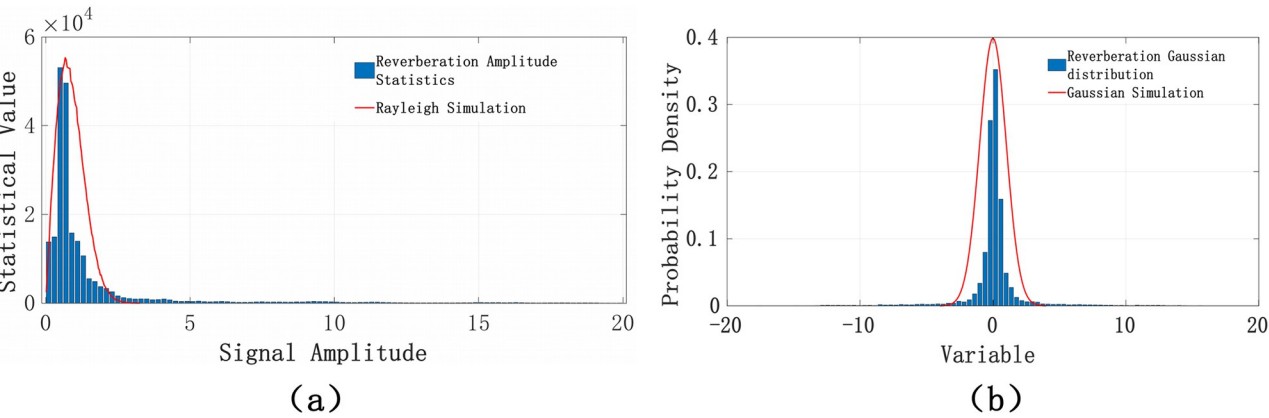

**Fig 3. Simulation results.**

The number of scatterers is assumed here. It is assumed that the number of scatterers generating reverberation on the $i$ ring is $N$, and the scattering characteristic function $P_i(t)$ at time $t$ is:

$$P_i(t) = \sum_{n=1}^{N} \frac{A}{r} e^{-jkr} R_{in} \frac{1}{r} e^{-ikr} \tag{1}$$

Where the signal strength is $A$, the $MA$ length is $r$, and the wave number is $k$, where $k = 2\pi f/c$, $r$ is the homeopathic frequency, the ocean sound speed is $c$, and the scattering coefficient of the $n$ scatterer is $R_{in} = a_{in}e^{j\psi_{in}}$, $1 \leq n \leq N$. The reverberation at time t can be expressed as:

$$R(t) = s(t) \otimes P_i(t) \tag{2}$$

In Eq 2, the emitted signal is $s(t)$, the reverberation signal is $R(t)$.

In Fig 3, the compliance of the generated signal is demonstrated. The envelope of the signal obtained by simulation will conform to the Rayleigh distribution as shown in Fig 3(a), and the data distribution will conform to the Gaussian distribution as shown in Fig 3(b).

The simulation of the reflected signal of the target body is added to the reverberation, and it is assumed that the sonar has $[1 \cdots m \cdots n]$ array of elements received [22]. The target echo signal received by the $m$ array element can be expressed as:

$$x_m(t) = \sum_{i=1}^{P} b_i S(t - \tau_{mi}) exp[W_{mi}(t) + 2\pi f_d(t) \cdot (t - \tau_{mi}) - \psi_i] \tag{3}$$

where the $W_{mi}(t)$ expression:

$$W_{mi}(t) = 2\pi f_0(t - \tau_{mi}) - \frac{2\pi}{\lambda}(r_m \cdot e_i) - \pi B(t - \tau_{mi}) + \frac{\pi B}{t}(t - \tau_{mi})^2 \tag{4}$$

In Eqs 3 and 4, the coordinate of the $m$ array element $(x_m, y_m, z_m)$ is represented by a vector $r_m$. The reference point $O(0, 0, 0)$ of the receiving array element is a point (the geometric center of the receiving array); $p$ represents the number of bright spots of the target; $b_i$ represents the reflection coefficient of the $i$-th bright spot; $S(t)$ represent the envelope of the transmitted signal; $\tau_{mi}$ represents the time delay experienced by the sound wave incident on the $i$ bright spot and then reflected the $m$ array element; $W_{mi}(t)$ represents the angular frequency change of the sound wave irradiated to the $i$ bright spot and then reflected to the $m$ array element; $f_d(t)$

represents the Doppler shift; $\psi_i$ represents the random phase shift of the $i$ two-point echo, uniformly distributed between $(0 \sim 2\pi)$. $B$ is the frequency modulation width ($B = 0$ is the CW signal).$\tau_{mi}$ in Eqs 3 and 4 is expressed in detail as:

$$\tau_{mi} = \frac{2r_i}{c} - \Delta\tau_{mi} \tag{5}$$

$r_i$ represents the distance from the $i$ th bright spot to the matrix reference point; $c$ is the speed at which sound waves travel through water. $\Delta\tau_{mi}$ represents the delay of the plane wave from the array element $m$ to the reference point of the array. The above method will realize the construction of a bright spot echo model.

In environmental noise simulation, the Marine environment has complex spatial and physical characteristics, and the noise level depends on mixing multiple noise sources. Due to the spatiotemporal variability of ambient noise, it is not easy to describe it accurately. Megan Liu's team and Bosheng Liu et al., based on a large number of measured data and data statistics on noise in shallow sea environment [23, 24], analyzed and obtained the empirical equation of shallow sea noise spectrum:

$$NL(f) = 10 lg f^{-1.7} + 6S + 55 \tag{6}$$

$NL$ represents noise spectrum level, $f$ represents frequency, and $S$ represents sea state level. At a certain depth, the amplitude distribution of environmental noise meets the Gaussian distribution. Therefore, after calculating the spectrum of Marine environmental noise, Gaussian noise can be used to approximate and simulate the time domain signal, generate a random noise sequence of $N(\cdot)$ and exchange its frequency domain information by Fourier transformation, and obtain the frequency domain function $R(\cdot)$ of random noise. Then the inverse Fourier transform of $N(f) = NL(f) \times R(f)$is performed to obtain the time domain Marine ambient noise sequence $N(t)$, that is the simulated Marine ambient noise sequence with specific amplitude distribution and spectral level curve requirements.

A series of factors, such as reverberation interference, environmental noise, bright spot model, and Doppler compensation, are comprehensively considered, and each part is simulated, and each element is superimposed to obtain the target echo signal, as shown in Fig 4:

## Signal processing

Some processing is required for the reverberation signals before the network training is performed. The generated signals must be processed in three steps: randomization of the target echo signal, normalization of the input data, partitioning the training set, validation set, and test set in a specific ratio. Some processing can make the network training converge faster and improve accuracy.

(1).Random generation is required for the echo signal strength and position, and only the diversity of input features can ensure the strong generalization ability of the generated model. According to the sonar equation, the loss during the propagation of the active sonar signal can be written as Eq 7.

$$2TL = SL - (NL - DI + DT) + TS \tag{7}$$

$TL$ represents the propagation loss because the active sonar is bidirectional, so $2TL$ defines its complete propagation loss; $SL$ represents the sound source level; $NL$ represents the noise level; $DI$ represents the directivity index; $TS$ represents the target strength. The propagation loss calculation formula can also be converted into the following distance-related empirical

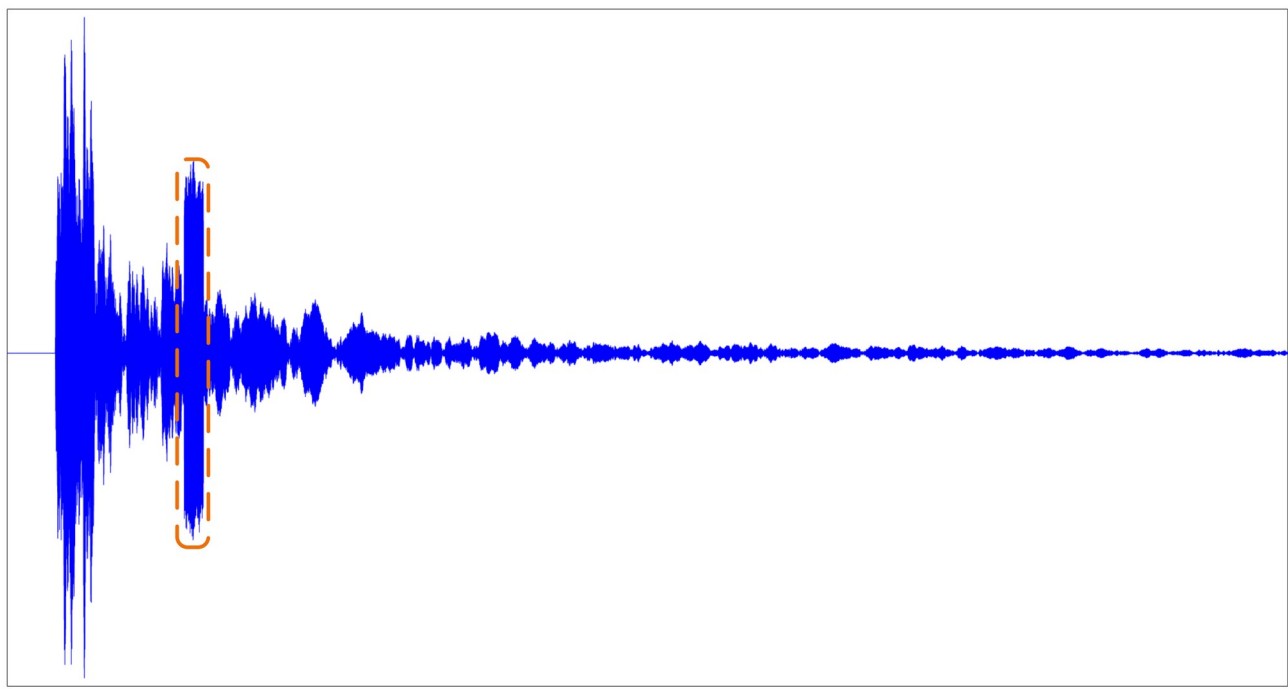

**Fig 4. Active sonar echo simulation.**

formula:

$$TL = 16 \times lg(r/1000) + 0.036 \times f^{3/2} \times (r/1000) + 60 \tag{8}$$

In Eq 8, $f$ represents the frequency, and $r$ represents the target distance. Substituting it into Eq 7, the variation of the target reflection intensity concerning the motion position can be realized. As shown in Eq 9:

$$S(t) = R(t) + 10^{TL/5} \times x_m(t) \tag{9}$$

(2).Normalize the data. The preprocessed data is constrained within a specific range, eliminating the adverse effects of individual sample data, and the input signal data has a standard dynamically adjusted range. Normalizing the data can accelerate the speed of gradient descent in finding the optimal solution and may improve accuracy.

$$S(t)' = \frac{S(t) - min(S)}{max(S) - min(S)} \times (L_1 - L_2) + L_2 \tag{10}$$

Where $S(t)'$ is the data after normalization, and normalize $S(t)$ to be between $[L_1, L_2]$.

(3).The reverberation data set is divided reasonably, and the entire data is divided into three sets: training set, validation set, and test set. The $S(t)'$ data is divided into a training set, validation set, and test set according to the ratio of 80%, 10%, and 10%. By setting the verification and test sets, the feature distribution is close to the training set, the analogy is compact, and the accuracy or loss is closer to reality.

## RS-U-Network construction

The sonar echo data after section 1.2 processing is $[m, n]$, which means that there are $m$ pieces of test data, and each amount of test data has $n$ points, which is represented as $[s^1, \ldots, s^n]$, where $s^i \in [-1, 1]$ for $i \in \{1, \ldots, n\}$. Here, we input the data into the neural network structure and perform feature extraction on the data, as shown in Fig 5.

The input time domain signal data, $S$, has a $n \times 147556$ size. Similarly, the output data, $S'$, also $n \times 147556$ size. For n sonar signals to be suppressed, a one-dimensional convolution of size 1 with $n \times m$ filters is used for feature extraction, filling zeros before the convolution to convert the feature stack of each sonar signal into a source prediction for each signal. By tanh nonlinearity, obtain source signal estimates with values in the interval $(-1, 1)$. In addition to upsampling, data concatenation is used to preserve the original features. Thus, the feature extraction and amplification of the initial sonar signal can be realized. This chapter introduces reverberation suppression networks in four parts. The first part uses 1D-CNN to extract time domain signal features, the second part is 1D Network Blocks, the third part is the realization of data splicing in the network structure, and the fourth part will introduce the realization of RS-U-Net model in detail.

## Feature extraction by 1D-CNN

In recent years, with the development of computer hardware, CNN has gradually been applied in various fields. 1D-CNN is very effective in processing time series data [25]. One-dimensional convolution is chosen to process signal data rather than two-dimensional convolution because one-dimensional convolution is more suitable for processing time series data, has higher computational efficiency, and can retain time information and extract timing features [26]. In this paper, 1D-CNN is used to extract and fuse the features of the input sonar signals, as shown in Fig 6.

These convolution kernels slide along the time axis of the sonar signal to extract more abstract and representative features from the sonar signal. The extracted features are combined to obtain a new sequence feature as the input of the next CNN layer. *Convolution kernel size =* [*Sampling rate/frequency*],where [·] indicates round down.The convolution kernel's size equals the number of points contained in a sinusoidal signal. As shown in Fig 6, four values can represent a sine wave, so the kernel size is set to 4. Convolution kernel size equal to a sine wave can help us analyze the periodic features in the signal.

Further, by taking the current moment in Fig 6 as an example, the convolution operation process could be expressed as "*feature sequence ⊗ convolution kernel*," where "⊗" represents

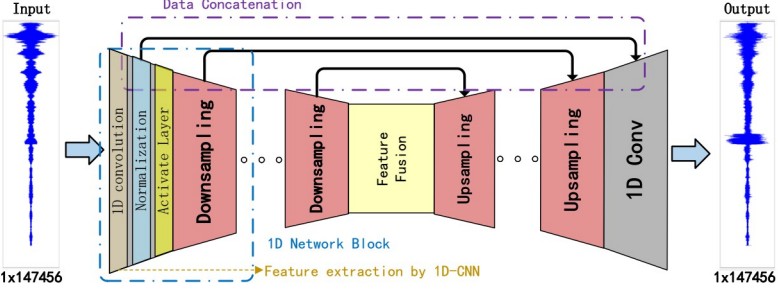

**Fig 5. Structure of RS-U-Net.**

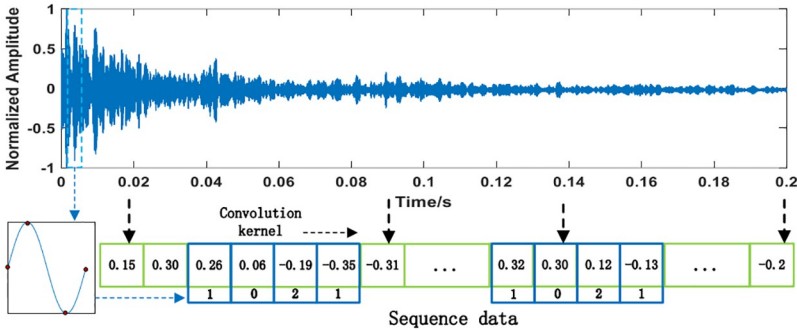

**Fig 6. Feature extraction sequence of 1D-CNN.**

the convolution operation symbol. Therefore, after the convolution operation, the number of extracted features was -0.47.

One-dimensional convolution is used to process time domain signals and can be used to extract local features from time series data using the sliding window. This sliding window captures current patterns and trends in the time series to better understand how the data changes. 1D-CNN When dealing with time series, the same convolution check can perform convolution operations on the entire sequence. This way of parameter sharing can significantly reduce the number of parameters in the model, improve computational efficiency, and help prevent overfitting. 1D-CNN has translation invariance; for translation operations in a time series, the result of the convolution operation does not change [27].

## 1D network block

This paper uses two types of blocks, the downsampling block and the upsampling block, to obtain better sonar signal $\mathbf{S}^{(n)}$ features. Since the two blocks are of the same form, only the positions of the upsampling and downsampling are different. In Fig 7, only the downsampling block is represented.

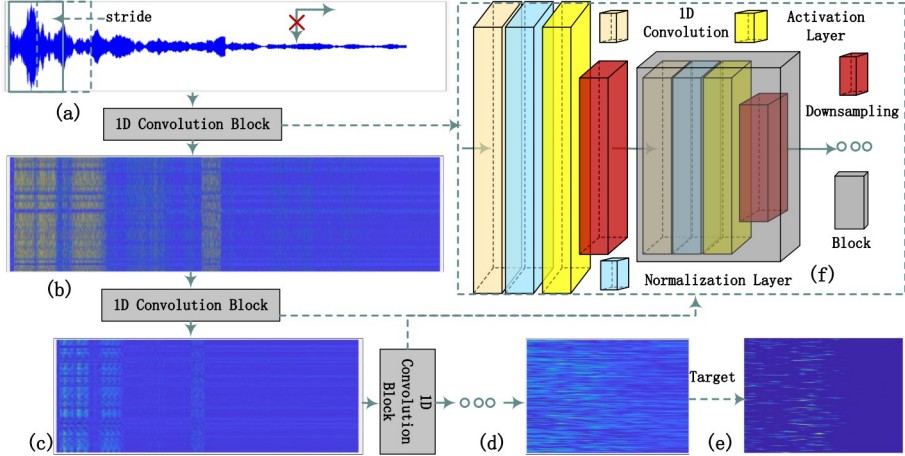

**Fig 7. Network block.**

In Fig 7, a network block consists of a convolution layer, a normalization layer, an activation layer, and a downsampling process. As shown in Fig 7(a) and 7(e), after one-dimensional convolution processing (data convolution dimension is 1), the normalization layer is used to re-normalize the data, and then the activation layer is used to process the results. The result after convolution is shown in Fig 7(b). For places with strong signals, the representation of features in the array after convolution is still strong, such as reverberation and target. Use downsampling to change the data length from $m$ to $m/2$. (For up-sampling, use linear interpolation to change the original length m to $m$ to $2^{*}m$.) The purpose of downsampling is to reduce the amount of computation, prevent overfitting, and increase the acceptable field so that the subsequent convolution kernel can learn more global information. After processing, Fig 7(c) is generated, the scale of the array is reduced, and more advanced features are extracted. The form shown in Fig 7(d) is formed after multiple convolutions. After several training and iterations, the feature map in Fig 7(e) formed by the final convolution is more prominent in the target's location, and the other features of the reverberation part disappear or are suppressed.

The diagram above shows the encoder structure. When the structure becomes a decoder, the convolution is transformed into a transposed convolution, and downsampling becomes an upsampling process.

## Data concatenation

Extracting sonar signal features is challenging. It is crucial to design a deep network structure to obtain more valuable recognition features from the dataset. However, as the network layers accumulate, training deep networks becomes a labour-intensive task due to a common insurmountable problem. This problem can be addressed more effectively by incorporating the UNet [28] approach, which optimizes data concatenation when transferring network parameters. Fig 8 in the paper illustrates the implementation of this approach.

The input sonar signal feature $x$ is propagated from the preceding layer and passed through a series of $Conv1D_1 \ldots Conv1D_n$ layers. The resulting data $x_n$ is obtained. $Conv1D_y$ processes $x$ separately through an alternative pathway, producing the output $Conv1D_y(x)$. The output $x_n \oplus Conv1D_y(x)$, known as $Concat(x)$, is then obtained. This operation combines the current features with the local ones, ensuring the original ones are preserved after multiple convolution

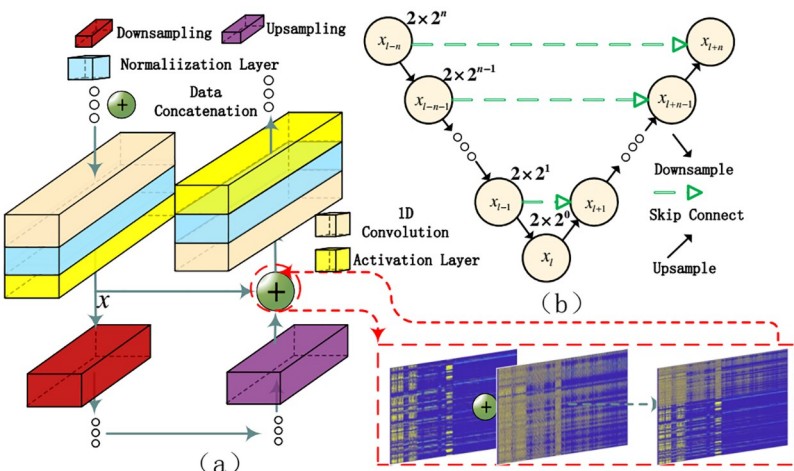

**Fig 8. Data concatenation.**

operations. The Decimate operation removes certain features at alternate time steps, reducing the time resolution by half. The Upsample operation increases the time resolution by two-fold using linear interpolation. The *Concat(x)* operation concatenates the current high-level features with the corresponding local features x. The convolution operation in the alternative pathway processes the data and generates the final output data. In Fig 8(a), the data concatenation operation is equivalent to the $\oplus$ operation, representing the skip connection shown in Fig 8(b).

The following network construction will be built with network blocks. The U-net network structure has the following characteristics. The network has $l + n$ layers, and each layer is labeled $[1, \ldots, l-1, l, l+1, \ldots, l+n]$, where $l - n = 1$.

For stacked layers, when the input is x, the learned feature is recorded as $\mathbf{H}(x_{l-n})$(echo signal feature). When $l = 0$, the cumulative layer only performs unit mapping at this time. Network performance does not degrade, allowing the accumulation layer to learn new features based on the input features, resulting in better performance.

A convolutional block of length $l + n$ can be expressed as:

$$\begin{cases} x_{l+n} = F_d(x_{l-n-1}, w_{l-n-1}), & n < 1 \\ x_{l+n} = F_u(x_{l+n-1}, w_{l+n-1}), & n > 1 \\ x_l = F_m(x_l, w_l), & n = 1 \end{cases} \tag{11}$$

$$x_{l+n} = F(x_{l+n}, w_{l+n}) \oplus H(x_{l-n}) \tag{12}$$

The convolution result can be expressed as *x* from *k* bocks and *y* block by continuous convolution iteration, where $l + n$ from $l - n - 1$ to $l + n - 1$. The input and output of the *l* residual unit of the formula are respectively represented in $x_l$, and each residual unit generally contains a multi-layer structure. $F_d$ is the downsampling block function, $F_u$ is the upsampling block function, representing the learned residual, and $h(x_l) = x_l$ depicts the identity mapping. The learned features from shallow layer *l* to deep layer *L* are expressed as:

$$x_L = x_l + \sum_{i=1}^{L-1} F(x_i, W_i) \tag{13}$$

The network is symmetrical, as depicted in Fig 8(b). The first half employs downsampling, while the second half employs upsampling. The network's architecture impacts the data processing length and the minimum distance for processing sonar signals, where $d = 2 \times 2^n$. When the downsampling block has *n* layers, the number of input points is at least 9. A symmetric network structure with 9 layers is used in the experiments, resulting in a minimum of $2^9 = 512$ input signal points. However, if only signal data with a length of 2048 is input, it will output only 1 value after 9 downsampling operations, leading to a diminished feature representation. To ensure adequate feature representation, the data signal length for training should be at least $d = 2 \times 512 = 1024$. The shortest detection distance for convolution is denoted as *L* and can be computed using the following formula.

$$L = \frac{d}{2 \times Fs} \times c \tag{14}$$

It can be known from the calculation that the detection distance of the processed active sonar is $3.072m$ in the network built by the 9-layer downsampling block.

## The RS-U-Net model

RS-U-Net refers to a modification of the U-shaped network. It turns its most classic two-dimensional convolution into a 1D-CNN that convolves specifically on signals, adding skip connections on their original basis to make it more accurate for signal feature extraction, as shown in Fig 9.

Fig 9 mainly shows its network structure's propagation process and main characteristics. The signal data S is directly input to the encoder layer $X_{En}^1$, and the one-dimensional convolution operation is started. The RS-U-Net is specifically designed to handle sonar signals. The role of the encoder is to convert the input sequence into a low-dimensional representation capable of capturing the key features of the input sequence. The decoder converts the encoding vector into a target sequence and dynamically generates content related to the target, as shown in $X_{De}^3$. The decoder directly receives feature maps from the scale encoder layer $X_{En}^3$. Its data size remains unchanged at $96 \times 768$. As the number of convolutional layers increases, the convolution of multiple neural networks may weaken data features, so the data crop structure is adopted to reduce the loss of information, as shown in Fig 8. The convolution of multiple neural networks has the following problems. As the number of convolution layers increases, the data features will be weakened, so the data clipping structure is used to reduce its partial disappearance. The significance of using one-dimensional convolution here is that its convolution direction is one-dimensional. This method is suitable for signal processing and feature extraction.

In RS-U-Net, the convolution with a stride of 1 maintains the output length equal to the input length. A downsampling method is employed to increase the receptive field of the original data by [1/2]. The signal data $\mathbf{S}^{L_m \times B} = [s^1, s^2, s^3, s^4, \cdots, s^{m-3}, s^{m-2}, s^{m-1}, s^m]$ is downsampled to $[s^1, s^3, \cdots, s^{n-2}, s^n]$. After convolving the data to obtain its minimum scale, corresponding upsampling [×2] is used along with interpolation to restore the data to its original scale. The signal is transformed from $S_n$ to $S_n'$ after processing, while the signal length remains unchanged.

Regarding physical structure, the target echo signal may contain multiple highlight echoes so that the target echo signal might be a multi-component signal. WVD suffers from severe cross-term interference when dealing with multi-component signals. Cohe's time-frequency

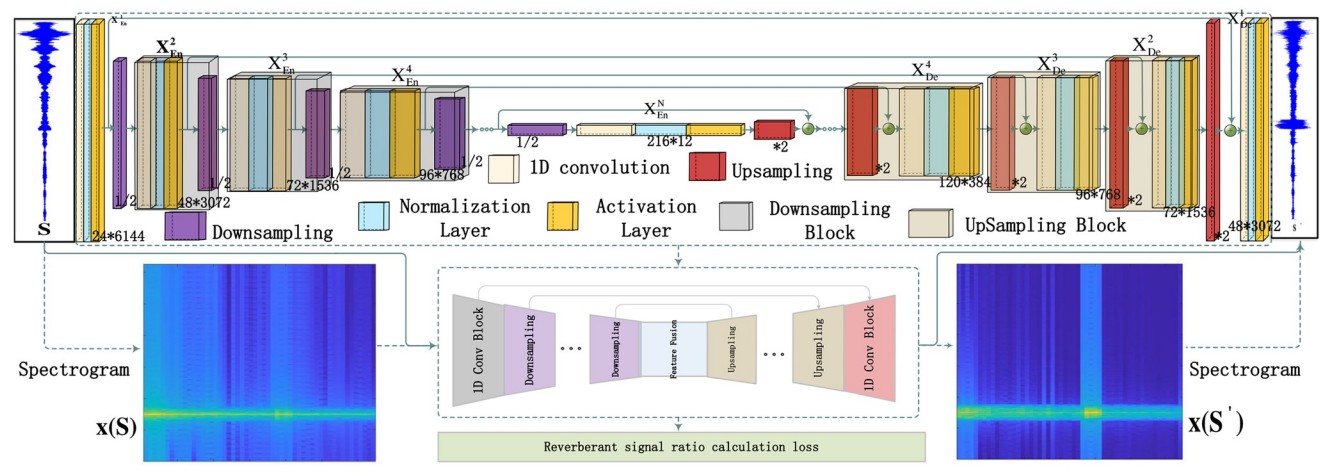

**Fig 9. RS-U-Net.**

distribution reduces the interference of cross terms to a certain extent by adding kernel functions, but its applicability to different signals has significant differences. Therefore, STFT is chosen as the joint time-frequency domain processing method for feature extraction of the observed $x(t)$. The RS-U-Net target will recover the clean target echo signal $x(S)$ in the reverberation observation $x(S')$.

## Experimental

In this section, the pool experiment data comparison demonstrates the RS-U-Net's ability to effectively reduce the active sonar signal's reverberation component. The proposed method's effectiveness and accuracy are shown by comparing various network structures. Additionally, we compared the proposed reverberation suppression method with other excellent methods to demonstrate its effectiveness.

### Experimental setup

This section describes the setup of the experiment. Experimental data were mainly obtained in the anechoic pool. The leading equipment of the experiment was active sonar equipment and cavity target object, and the main equipment of the experiment is shown in Fig 10 below.

A sonar detection system is used to acquire sonar signals with reverberation, as shown in Fig 10(a). The sonar device is an irregular array composed of 30 oscillators, and the signal is transmitted as a 30 kHz continuous wave sinusoidal signal. The sampling rate of the signal echo is 250kHz, and the default underwater sound velocity during calculation is 1500m/s. The experimental target of the equipment is shown in Fig 10(b). The experimental target is a cylindrical cavity with a diameter of 533 mm and a length of 1.5m. The sonar detection equipment and the target are mounted on the driving and rotating experimental platform for easy movement and rotation.

The active sonar equipment was fixed during the experiment, and the target was moved. The main experimental methods are shown in Fig 11.

The experiment to obtain experimental data was mainly conducted in a pool with an acoustic wedge of $13m \times 8m \times 8m$. The acoustic wedge can reduce the acoustic signal reflection of the water bottom and the surrounding pool wall. During the experiment, the sonar

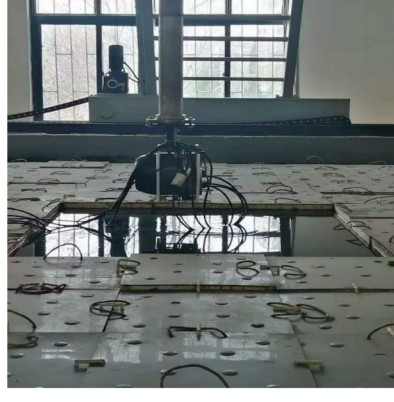 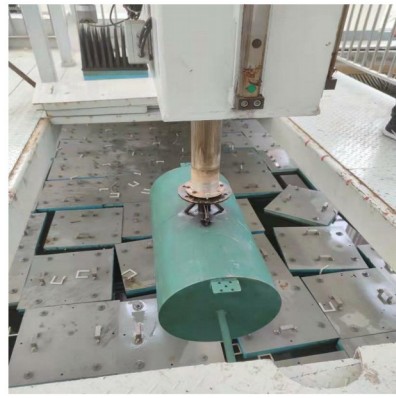

（a）                              （b）

**Fig 10. Experimental equipment.**

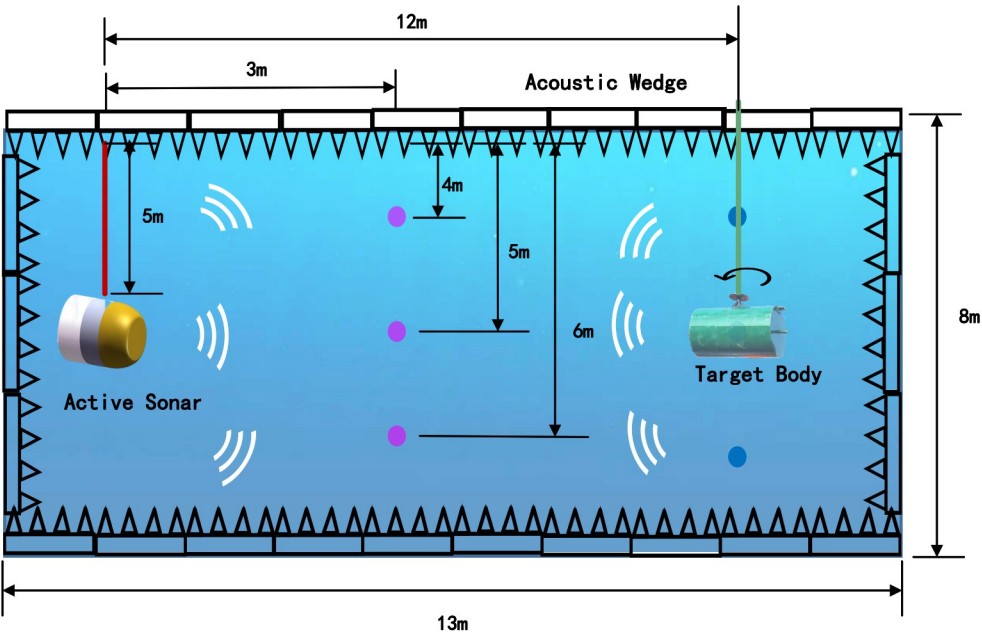

**Fig 11. Equipment and target experimental setup.**

detection equipment was fixed at a depth of 5m underwater, and the target object was moved between 3 and 12m away from the sonar equipment. It was placed at a depth of 4m, 5m, and 6m, respectively, for the experiment. Signals from different angles were collected, and the target object was rotated to simulate echoes of different intensities. When the smaller surface faces the sonar device, a smaller intensity echo signal can be obtained; otherwise, a stronger echo signal will be obtained. Detailed experiments and equipment are shown in Fig 12.

During the experiment. In contrast, a depth adjustment experiment is carried out, the translation distance of the equipment will also be tested, and the translation distance is about 1m about the position of the target body directly opposite. During the experiment, DC powers were used to power the sonar equipment. The current clamp was used to judge whether the piezoelectric ceramic module in the sonar equipment was usually started, and the spectrograph displayed the results. In the experiment, the start of the computer sonar equipment and the acquisition of signals are controlled. At the same time, another computer is connected to the hydrophone to monitor the signal sent by the sonar equipment to see whether the signal generated by the experiment can meet the requirements of the experiment.

During the experiment, nine points were measured at different distances, and different reflecting surfaces were tested at different points. The setup of the experiment is described in detail above. Through the above experimental Settings, the effectiveness and credibility of the method for different data are verified.

## Experiment verification

In this section, the method proposed in this paper will be applied to the data obtained from the above experiments to verify the effectiveness and advancement of the method. We will use SRR to indicate the degree to which the signal is suppressed and use it as a standard for

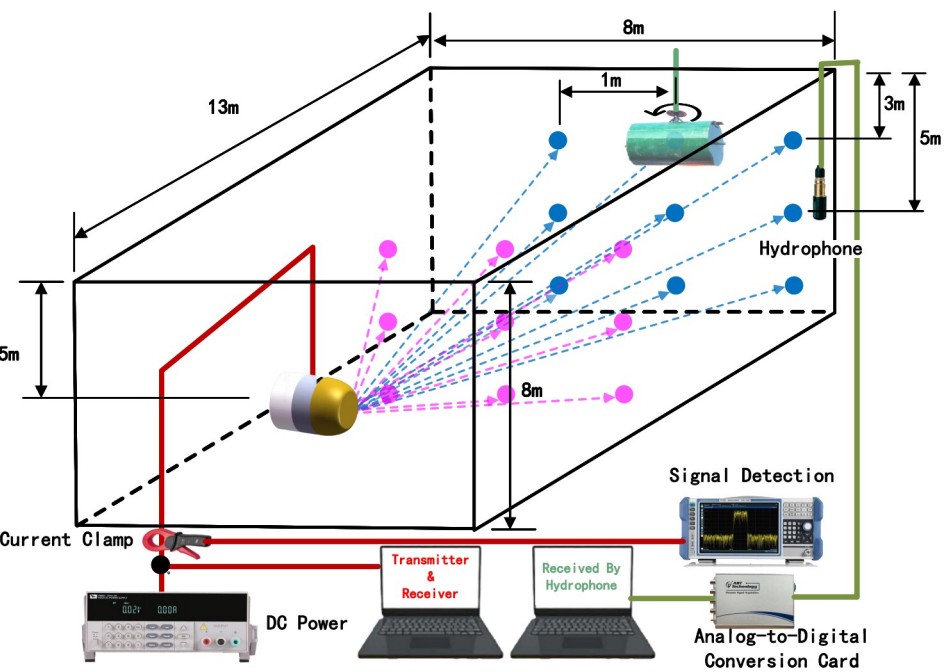

**Fig 12. Detailed experimental setup.**

comparison with other methods. The SRR formula is expressed as follows:

$$SRR = 10lg(\frac{\mathbf{P}_{Signal} - \mathbf{P}_{reverberation}}{\mathbf{P}_{reverberation}}) \qquad (15)$$

In Eq 15, $\mathbf{P}_{Signal}$ is the power of the activate sonar signal, the $\mathbf{P}_{reverberation}$ is the power of the reverberation signal, and $\mathbf{P}_{Signal} - \mathbf{P}_{reverberation}$ is the power of the echo signal. SRR can be used to indicate the degree of reverberation suppression.

The comparison experiment is carried out by setting parameters to verify that the selected parameters are the best, as shown in Table 1.

For different network structures, the ability of signal features is other for learning. Network structures are not only more profound, the better. Here, the network structures of 7, 8, and 9 layers were tested for comparison. At the same time, the loss is compared in two ways: mean square error (MSE) and mean absolute error (MAE). Upsampling is compared in two ways: linear interpolation and transposed convolution. Weight initialization is compared in yes and no. The loss comparison diagram is shown in Fig 13.

**Table 1. Comparison of different parameter settings.**

| Models Name | Layers | Loss | Upsampling | Weight Initialization |
|:---:|:---:|:---:|:---:|:---:|
| Model 1 | 8 | MSE | Linear Interpolation | No |
| Model 2 | 9 | MSE | Linear Interpolation | No |
| Model 3 | 9 | MAE | Transposed Convolution | No |
| Model 4 | 9 | MSE | Linear Interpolation | Yes |
| Model 5 | 10 | MSE | Linear Interpolation | No |

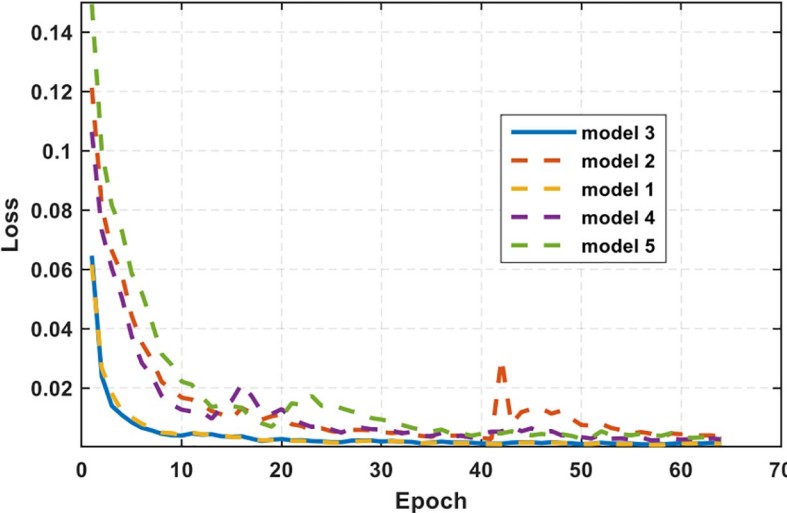

**Fig 13. Loss curve.**

In Fig 13, the loss represents the error between the effect generated by the network and the real expected. After comparative experiments, it is shown that although model 1 is similar to model 3, its convergence speed is still weaker than when the network depth is 9 layers, its convergence speed is the fastest, and its loss is the smallest. The experiment's results continued to converge, reaching around 0.001, and no longer changed.

Fig 14 shows the results of the influence of different network parameter Settings on the degree of reverberation suppression during training. As the number of training increases, the SRR of the validation set data increases from 6dB to 29dB, which is about 23dB better than the signal-to-reverberation ratio before processing. Different models have different processing

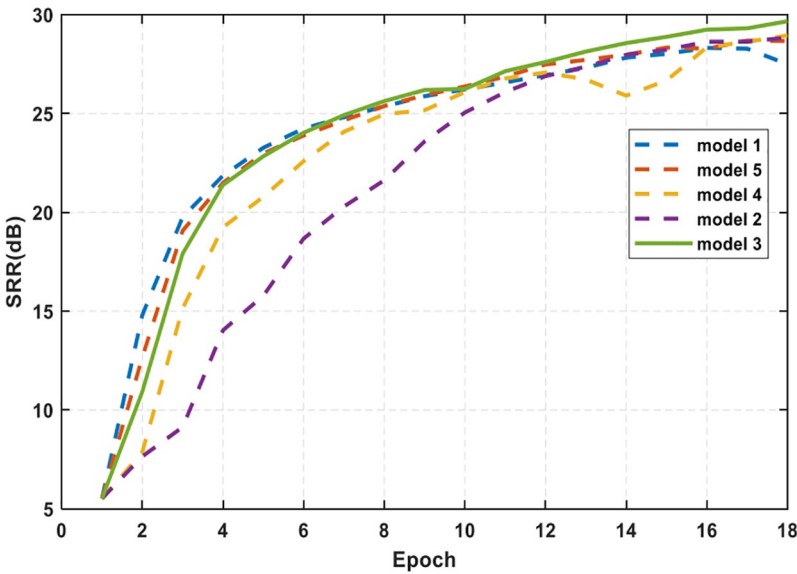

**Fig 14. SRR comparison.**

capabilities for underwater acoustic signals. When the number of layers is 8, the reverberation suppression efficiency is faster, and its reverberation suppression ability is reflected in the SRR. The final reverberation suppression effect is about 3dB higher than the other methods.

The active sonar of the signal is subjected to reverberation suppression, and the process of the signal during training is shown in Fig 15.

Fig 15(a) represents the original signal; it can be seen from the time-domain diagram that the target is not evident in the reverberation background, and it can be seen from the spectrum diagram that the target echo frequency is 30kHz. The position of the target echo cannot be seen from the time-frequency diagram. In Fig 15(b)–15(e) are the results of the network's continuous learning to suppress the signal, corresponding to the results after epoch 20, epoch 40, epoch 60, and epoch 80, respectively. The figure shows the comparison of different stages. The first is the comparison of the signal time domain diagram, the second is the echo signal frequency domain diagram, and the third is the comparison of the signal time-frequency diagram. Each row is rendered differently for the current state. In Fig 15(a), the original active sonar signal echo is initially submerged in the superimposed signal of reverberation and noise, which is difficult to show through the frequency domain and time-frequency diagram, as shown in Fig 15. The process is shown in (b) and (c). After extracting features through RS-U-Net, the information is constantly corrected and iterated. The echo signal needs to be cleaner, and the 30kHz echo is masked. The target can be found on the time-frequency plot after the

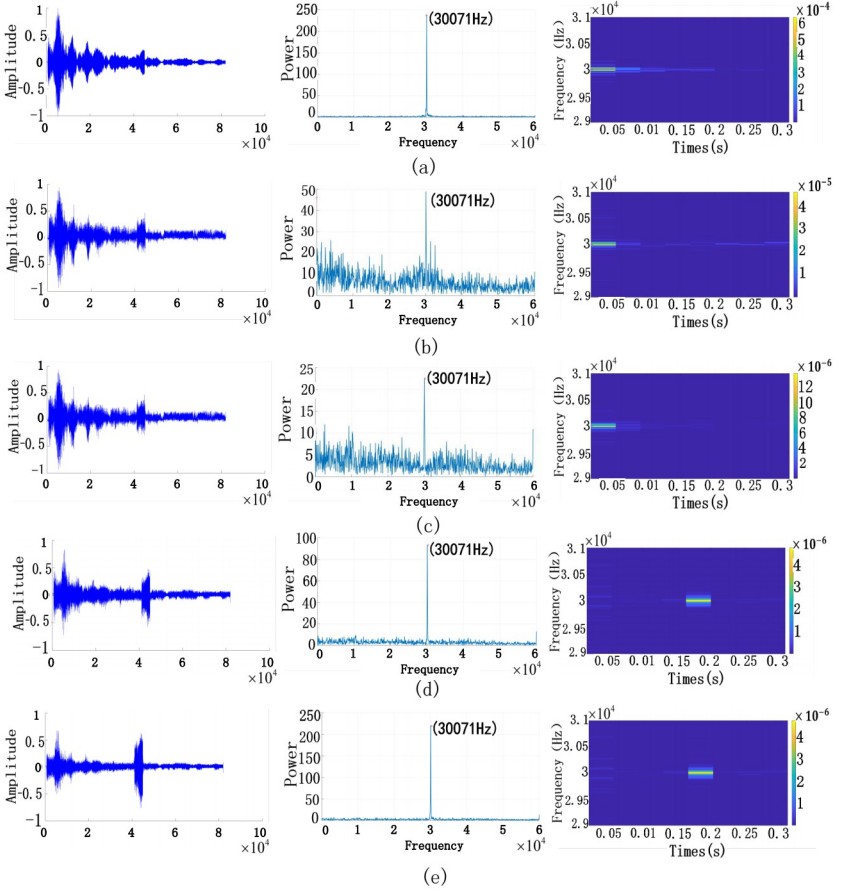

**Fig 15. Active sonar reverberation suppression change graph.**

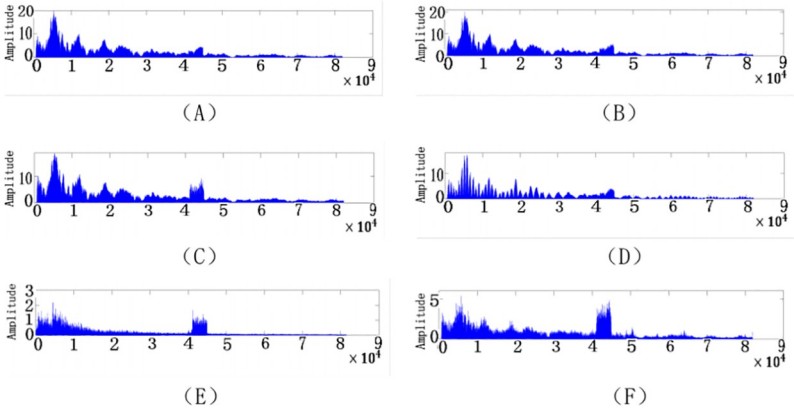

**Fig 16. Comparison of classical methods.**

final iteration, as shown in Fig 15(d) and 15(e). When it arrives in Fig 15(e), the echo is already a 30kHz signal. On the time-frequency spectrum, the echo shows strong signal characteristics, and the position of the echo signal can also be seen through the time-frequency spectrum.

The comparison effect display diagram is shown in Fig 16. Here, we compare various excellent classical reverberation suppression methods to demonstrate the effectiveness of RS-U-Net in suppressing reverberation.

Fig 16(A) is the original data in the comparison chart. It is impossible to see the target echo position very clearly. The target echo is masked. Fig 16(B) is a diagram showing the effect of the autoregressive pre-whitener [29] reverberation suppression method. Fig 16(C) shows the impact of the least mean square filter (LMS) [30]. It found that the reverberation part is suppressed and reduced amplitude. Fig 16(D) is a diagram showing the effect of reverberation suppression by adaptive fractional Fourier transform(FrFt) [31]. Fig 16(E) shows the impact of Principal Component Inversion (PCI) reverberation suppression [32]. Fig 16(F) shows the impact of the method RS-U-Net proposed in this paper. Compared to the excellent method, the reverberation part is effectively suppressed, and the echo part is displayed.

The superiority of the method proposed in this paper can be seen through the comparison chart, and its specific parameters are shown in Table 2. The comparison of the parameter model size, reverberation power, echo power, SRR, and improved SRR between the existing algorithm and the proposed method is given in the table.

**Table 2. Experimental data comparison table.**

| Algorithms | Params(M) | Reverberation Power | Echo Power | SRR(dB) | Imporve SRR(dB) |
|---|---|---|---|---|---|
| Original Data | — | 33.67 | 21.02 | -2.05 | 0 |
| Ar PreWhitener | — | 33.65 | 21.07 | -2.03 | 0.02 |
| LMS | — | 9.1 | 6.62 | -1.38 | 0.67 |
| FrFt | — | 10.94 | 15.06 | 1.38 | 3.43 |
| PCI | — | 6.03 | 28.34 | 6.72 | 8.77 |
| PCI-SVM | — | 6.15 | 29.17 | 6.76 | 8.81 |
| Wave-U-Net | 46.8 | 5.44 | 17.64 | 5.11 | 7.16 |
| SEW-U-Net | 50.4 | 2.16 | 11.95 | 7.43 | 9.48 |
| RS-U-Net | **38.7** | 2.77 | 17.74 | 8.06 | **10.11** |

In table algorithm comparison, the parameter Params represents the size of the parameter model, and a smaller parameter model means that its processing speed is faster. Reverberation Power represents the strength of Reverberation part, and Echo power represents the strength of the Echo part. SRR can be obtained according to the calculation of Reverberation Power and Echo power(Eq 15). Improve SRR is obtained by comparing SRR with original data. The parameters on Imporve SRR can compare the effectiveness of both methods on reverberation suppression.

Compared with classical algorithms, Ar PreWhitener and LMS have low improvement on SRR, which are below 1dB. FrFt can improve by 3.43 dB, but the effect is still not as good as PCI. PCI and PCI-SVM [33] methods are the best among the comparison methods, which increase SRR from -2.05dB to about 6.7dB and suppress reverberation signal from 33.67 dB to about 6. By interpreting the algorithm, the PCI-SVM method increases the adaptive selection of rank, which can select the appropriate rank according to the signal. After many experiments, the effect is best when the rank is 8. Compared with other U-shaped networks, this paper chooses Wave-U-Net [34] and SEW-U-Net [35] for comparison, mainly applied and audio separation. Experimental verification shows that it also has a specific effect on reverberation suppression. It can be seen from Params that the parameter model size of RS-U-Net is 38.7M, which is better than other models.

Moreover, by improving the SRR comparison, the proposed method can improve the signal up to 10.11dB. The RS-U-Net method proposed in this paper can suppress reverberation and distinguish and reduce the suppression degree of the echo signal. Through SRR comparison, the improvement amplitude of the signal reverberation ratio of these methods is 0.02dB, 0.67dB, 3.43dB, 8.77dB, 8.81dB, 7.16dB, 9.48dB, and 10.11dB, respectively. The method proposed in this paper can effectively improve the signal-reverberation ratio and realize the suppression of reverberation. Compared with other methods, it has certain advantages.

Preliminary experiments have proved that the proposed method of reverberation suppression using RS-U-Net has certain superiority and rationality when constructing a 9-layer network encoder. The signal quality is improved compared with excellent algorithms. The signal quality is improved compared with excellent algorithms. The measured signal-to-reverberation ratio is improved by 10.11dB, proving the method's effectiveness and superiority in suppressing reverberation.

## Conclusion

Reverberation suppression is an essential issue in an active sonar system. This paper presents a method for reverberation suppression of sonar signals using an end-to-end neural network. The proposed network uses one-dimensional convolution to effectively suppress the processed sonar signal in this reverberation suppression network called RS-U-Net. The efficient suppression of RS-U-Net lies in its skip-connected network structure, and the signal features are no longer easily lost after multi-layer convolution. The efficiency of the method proposed in this paper can be proven in the comparison algorithm. The overall sonar reverberation signal can be suppressed, and its SRR can be improved by about 10 dB.

After many experiments, it has been found that RS-U-Net has unique requirements for the transmission pulse width. Different detection distances need to adjust the transmission pulse width of the signal, but this is also the pulse width of RS-U-Net for echo signals.,the width is not sensitive. Of course, this also has a particular relationship with the data set. In the following research, its generalization ability will be increased, and it can also have an efficient processing ability for signals with different pulse widths.

## Supporting information

**S1 File. All data files are available from the figshare database.** (https://doi.org/10.6084/m9.figshare.24151551).
(DOCX)

## Author Contributions

**Conceptualization:** Xiao Chen, Yuan An.

**Writing – original draft:** Zhen Wang.

**Writing – review & editing:** Hao Zhang, Xiao Chen, Yuan An.

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
