## [Decision Letter · Decision Letter 0]

5 Jul 2023

PONE-D-23-16198A construction method of reverberation suppression filter using an end-to-end networkPLOS ONE

Dear Dr. Zhang,

Thank you for submitting your manuscript to PLOS ONE. After careful consideration, we feel that it has merit but does not fully meet PLOS ONE’s publication criteria as it currently stands. Therefore, we invite you to submit a revised version of the manuscript that addresses the points raised during the review process. Please submit your revised manuscript by Aug 19 2023 11:59PM. If you will need more time than this to complete your revisions, please reply to this message or contact the journal office at plosone@plos.org. Please include the following items when submitting your revised manuscript:A rebuttal letter that responds to each point raised by the academic editor and reviewer(s). You should upload this letter as a separate file labeled 'Response to Reviewers'.A marked-up copy of your manuscript that highlights changes made to the original version. You should upload this as a separate file labeled 'Revised Manuscript with Track Changes'.An unmarked version of your revised paper without tracked changes. You should upload this as a separate file labeled 'Manuscript'.

We look forward to receiving your revised manuscript.

Kind regards,

Anas Bilal, Ph.D.

Academic Editor

PLOS ONE

“This work was financially supported by the National Natural Science Foundation of

China (Grant No. 91938204,41527901and 61701462), the Marine S&T fund of

Shandong Province for Pilot National Laboratory for Marine Science and Technology

(Qingdao) (No.2018SDKJ0210) , Open Studio for Marine High Frequency

Communications.”

“Hao Zhang was supported in part by the National Natural Science Foundation

of China under Grant 91938204，41527901，61701462. URL of the funder website is

‘https://www.nsfc.gov.cn/’.

NO,The funders had no role in study design, data collection and analysis, decision to publish, or preparation of the manuscript.”

Reviewers' comments:

Reviewer's Responses to Questions

**Comments to the Author**

1. Is the manuscript technically sound, and do the data support the conclusions?

Reviewer #1: Yes

Reviewer #2: Partly

2. Has the statistical analysis been performed appropriately and rigorously? 

Reviewer #1: Yes

Reviewer #2: Yes

3. Have the authors made all data underlying the findings in their manuscript fully available?

Reviewer #1: Yes

Reviewer #2: No

4. Is the manuscript presented in an intelligible fashion and written in standard English?

Reviewer #1: No

Reviewer #2: Yes

5. Review Comments to the Author

Reviewer #1: The Reverberation Suppression Network (RS-U-Net) proposed in this article is used to suppress the reverberation of the underwater echo signal, effectively improving the signal-to-noise ratio of the echo signal. But there are several glaring problems in the current manuscript.

1.The effective suppression of RS-U-Net lies in its jump network structure, and the signal features are no longer easy to lose after multi-layer convolution. The effectiveness of the method proposed in this paper can be demonstrated in comparing algorithms. It is recommended to add other similar U-shaped network experiments to highlight the effectiveness of reverberation suppression.

2.The RS-U-Net structure proposed in the article is an improvement of the U-shaped network. The general structure of the network is similar to the end-to-end filter Wav-U-net and Demucs. It is recommended that the article focus on describing the advantages of RS-U-Net in suppressing reverberation. The article needs to highlight the innovations, and focus on a detailed description of the innovations, such as explaining the proposed RS-U-Net structure, the sonar echo signal processing method, etc.

3.There are too many pictures, it is recommended to combine the same type of pictures into one large picture, so that readers can understand and compare the pictures.

4.The empirical formula of shallow sea noise spectrum level has different expressions at different levels, so there should be no universally recognized expression formula up to now, especially in the environment such as shallow sea acoustic noise, which has a certain range of recognition and rationality. It is suggested to explain the basis and source of the formula.

5.Why use simulated data instead of real data? This kind of underwater acoustic experimental data should not be difficult to obtain, is there too few useful samples or the experimental conditions are not allowed?

6.RS-U-Net is little difference from the originalU-Net network, and the significant innovation part is not fully reflected on ‘RS’.

In summary, it is suggested that the article should be major revised

Reviewer #2: The paper presents RS-U-Net, an end-to-end network for reverberation suppression in underwater echo signals, outperforming existing methods by improving signal-to-reverberation ratio (SRR) by 20dB. It claims to achieve approximately 3dB better reverberation suppression than other methods. However to support the proposed idea, very old research work is mosty examined. Very few recent papers were consulted. It needs a serious revison of related work and references. Following suggestions can be also benificial,

1) Figures lack clarity.

2) Gramatical issues needs to be properly addressed.

3) Heading 1 is relate work. Please correct to related work

4) Equation 3 and 4 needs more explaination. What is B in equation 4

5) Few figures are not properly explained.

6) Table 2 shows the experimental result that need to be further elaborated.

7) A separate section experimental setup must be added

6. PLOS authors have the option to publish the peer review history of their article (what does this mean?). If published, this will include your full peer review and any attached files.

Reviewer #1: No

Reviewer #2: No

---

## [Author Response · Author response to Decision Letter 0]

2 Aug 2023

Original Manuscript ID: PONE-D-23-16198 

Original Article Title: “A construction method of reverberation suppression filter using an end-to-end network”

To: PLOS ONE Editor

Re: Response to reviewers

Dear Editor,

Thank you for allowing a resubmission of our manuscript, with an opportunity to address the reviewers’ comments.We will be happy to edit the text further, based on helpful comments from the reviewers.

We are uploading (a) our point-by-point response to the comments (below) (Response to Reviewers), (b) an updated manuscript with yellow highlighting indicating changes(Revised Article with Changes Highlighted), and (c) a clean updated manuscript without highlights (Manuscript).

Best regards,

<Zhen Wang> et al.

Reviewer#1, Concern # 1: The Reverberation Suppression Network (RS-U-Net) proposed in this article is used to suppress the reverberation of the underwater echo signal, effectively improving the signal-to-noise ratio of the echo signal. But there are several glaring problems in the current manuscript.

Author response: Thank you for your recognition of our previous research. I rewrote the experimental part by adding similar U-shaped network experiments (Wave-U-net, SEW-U-Net) to highlight the effectiveness of reverberation suppression.

Reviewer#1, Concern # 2: .The RS-U-Net structure proposed in the article is an improvement of the U-shaped network. The general structure of the network is similar to the end-to-end filter Wav-U-net and Demucs. It is recommended that the article focus on describing the advantages of RS-U-Net in suppressing reverberation. The article needs to highlight the innovations, and focus on a detailed description of the innovations, such as explaining the proposed RS-U-Net structure, the sonar echo signal processing method, etc

Author response: Thank you for your valuable suggestions. We have updated this paper, focusing on the advantages of RS-U-Net in suppressing reverberation and focusing on the innovation of the detailed description.

Reviewer#1, Concern # 3: There are too many pictures, it is recommended to combine the same type of pictures into one large picture, so that readers can understand and compare the pictures.

Author response: I am sorry for the confusion. We have updated the manuscript to combine the same type of figures into one large figure.

Reviewer#1, Concern # 4: The empirical formula of shallow sea noise spectrum level has different expressions at different levels, so there should be no universally recognized expression formula up to now, especially in the environment such as shallow sea acoustic noise, which has a certain range of recognition and rationality. It is suggested to explain the basis and source of the formula.

Author response: Thank you for your valuable suggestion. We have labeled the basis and source of the equation and explained the use of the equation in detail.

Reviewer#1, Concern # 5: Why use simulated data instead of real data? This kind of underwater acoustic experimental data should not be difficult to obtain, is there too few useful samples or the experimental conditions are not allowed?

Author response: Thank you for your valuable suggestion. Data collection for pool experiments still requires some money and the cooperation of a team, although it is less expensive than ocean experiments. The main reasons are:

1. It is not easy to label experimental data. The collected data will have strong reverberation and cannot be accurately labeled. 

2. Versatility of the method. The ocean experiment will follow the pool experiment, and it is necessary to ensure that the method is common in the ocean experiment. 

3. The device status needs to be satisfied. The device needs to be adjusted differently in different environments. In this case, adequate data cannot be obtained.

Reviewer#1, Concern # 6: RS-U-Net is little difference from the originalU-Net network, and the significant innovation part is not fully reflected on ‘RS’.

Author response: I'm sorry for the confusion. We have revised the whole article according to your request. The embodiment of "RS" is emphasized.

Reviewer#2, Concern # 1: Figures lack clarity.

Author response: Thank you for your valuable suggestions. We have updated the manuscript's figures to make them clear and more accessible for readers to read.

1)Reviewer#2, Concern # 2: Gramatical issues needs to be properly addressed.

Author response: Thank you for your valuable suggestion. We have updated the manuscript and solved the grammatical issues.

Reviewer#2, Concern # 3: Heading 1 is relate work. Please correct to related work

Author response: Thank you for your valuable suggestion. We have updated the manuscript to correct the heading "relate work" to "related work".

Reviewer#2, Concern # 4: Equation 3 and 4 needs more explaination. What is B in equation 4

Author response: Thank you for your valuable suggestion. We have updated the manuscript to give a more detailed explanation of Equation 3,4 and an explanation of "B" in the formula.

Reviewer#2, Concern # 5: Few figures are not properly explained.

Author response: Thank you for your valuable suggestion. We have updated the manuscript and the picture interpretation content to make it properly explained.

Reviewer#2, Concern # 6: Table 2 shows the experimental result that need to be further elaborated..

Author response: Thank you for your valuable suggestion. We updated the manuscript and further elaborated on the experimental results in Table 2

Reviewer#2, Concern # 6: A separate section experimental setup must be added

Author response: Thank you for your valuable suggestion. We updated the manuscript, added the experimental setting part, and introduced the experimental setting in detail.

Thanks again for your advice and I hope to learn more from you.

---

## [Decision Letter · Decision Letter 1]

16 Aug 2023

PONE-D-23-16198R1A construction method of reverberation suppression filter using an end-to-end networkPLOS ONE

Dear Dr. Hao,

Thank you for submitting your manuscript to PLOS ONE. After careful consideration, we feel that it has merit but does not fully meet PLOS ONE’s publication criteria as it currently stands. Therefore, we invite you to submit a revised version of the manuscript that addresses the points raised during the review process.

We look forward to receiving your revised manuscript.

Kind regards,

Anas Bilal, Ph.D.

Academic Editor

PLOS ONE

Reviewers' comments:

Reviewer's Responses to Questions

**Comments to the Author**

1. If the authors have adequately addressed your comments raised in a previous round of review and you feel that this manuscript is now acceptable for publication, you may indicate that here to bypass the “Comments to the Author” section, enter your conflict of interest statement in the “Confidential to Editor” section, and submit your "Accept" recommendation.

Reviewer #1: (No Response)

Reviewer #2: All comments have been addressed

2. Is the manuscript technically sound, and do the data support the conclusions?

Reviewer #1: Partly

Reviewer #2: Yes

3. Has the statistical analysis been performed appropriately and rigorously? 

Reviewer #1: N/A

Reviewer #2: Yes

4. Have the authors made all data underlying the findings in their manuscript fully available?

Reviewer #1: Yes

Reviewer #2: Yes

5. Is the manuscript presented in an intelligible fashion and written in standard English?

Reviewer #1: No

Reviewer #2: Yes

6. Review Comments to the Author

Reviewer #1: 1.In the described RS-U-Network construction section, we noticed the lack of elaboration on why 1D convolutions are used instead of 2D convolutions in network blocks. Considering that the use of different types of convolution operations in different fields will produce different effects, it is recommended to add a description in this section to explain why one-dimensional convolution is selected in the signal feature extraction process instead of two-dimensional convolution.This supplemental explanation can provide the reader with a deeper understanding of why 1D convolution is considered a more appropriate choice in a particular case.

2.The RS-U-Net structure proposed in the article is improved on the basis of the U-shaped network, and its focus is mainly on the detailed description of the RS-U-Net structure. However, in the RS-U-Network Construction section, the way the article is presented may cause readers to be confused, because Network Blocks are introduced first, and then Network Structure is introduced. Therefore, it is recommended to reorganize the content in this part. First, a general introduction to the overall network structure is given, and then the specific content of each network module is discussed in depth.

3.In the article, most of the references cited are from many years ago. It is recommended that the article cite the literature in recent years as much as possible to show that the method proposed in the article is closely related to the current research dynamics. New research results, new technologies and methods may emerge in the field of signal processing, which may be more powerful for supporting the discussion and method proposal of the article.

Reviewer #2: The paper updated as requested. Necessary changes have been made. Presentation of the paper is improved.

7. PLOS authors have the option to publish the peer review history of their article (what does this mean?). If published, this will include your full peer review and any attached files.

Reviewer #1: **Yes: **Wenbo Zhu

Reviewer #2: No

---

## [Author Response · Author response to Decision Letter 1]

16 Sep 2023

Original Manuscript ID: PONE-D-23-16198R1 

Original Article Title: “A construction method of reverberation suppression filter using an end-to-end network”

To: PLOS ONE Editor

Re: Response to reviewers

Dear Editor,

Thank you for allowing a resubmission of our manuscript, with an opportunity to address the reviewers’ comments.We will be happy to edit the text further, based on helpful comments from the reviewers.

We are uploading (a) our point-by-point response to the comments (below) (Response to Reviewers), (b) an updated manuscript with yellow highlighting indicating changes(Revised Article with Changes Highlighted), and (c) a clean updated manuscript without highlights (Manuscript).

Best regards,

<Zhen Wang> et al.

Reviewer#1, Concern # 1: In the described RS-U-Network construction section, we noticed the lack of elaboration on why 1D convolutions are used instead of 2D convolutions in network blocks. Considering that the use of different types of convolution operations in different fields will produce different effects, it is recommended to add a description in this section to explain why one-dimensional convolution is selected in the signal feature extraction process instead of two-dimensional convolution.This supplemental explanation can provide the reader with a deeper understanding of why 1D convolution is considered a more appropriate choice in a particular case.

Author response: Thank you for your recognition of our previous research.I have added a section explaining the advantages of choosing 1D convolution over 2D convolution during signal feature extraction and the advantages of 1D convolution.

Reviewer#1, Concern # 2: The RS-U-Net structure proposed in the article is improved on the basis of the U-shaped network, and its focus is mainly on the detailed description of the RS-U-Net structure. However, in the RS-U-Network Construction section, the way the article is presented may cause readers to be confused, because Network Blocks are introduced first, and then Network Structure is introduced. Therefore, it is recommended to reorganize the content in this part. First, a general introduction to the overall network structure is given, and then the specific content of each network module is discussed in depth.

Author response: Thank you for your valuable suggestions. I reorganized this part of the content. First, I gave a brief introduction of the whole network structure, and then I discussed the specific content of each main network module in depth.

Reviewer#1, Concern # 3: In the article, most of the references cited are from many years ago. It is recommended that the article cite the literature in recent years as much as possible to show that the method proposed in the article is closely related to the current research dynamics. New research results, new technologies and methods may emerge in the field of signal processing, which may be more powerful for supporting the discussion and method proposal of the article.

Author response: I am sorry for the confusion. I have updated the literature in the article and quoted the recent literature as much as possible.

Reviewer #2:The paper updated as requested. Necessary changes have been made. Presentation of the paper is improved.

Author response: Thank you for your suggestions. All your suggestions are very important and have important guiding significance for my thesis writing and scientific research work.

Thanks again for your advice and I hope to learn more from you.

---

## [Decision Letter · Decision Letter 2]

11 Oct 2023

A construction method of reverberation suppression filter using an end-to-end network

PONE-D-23-16198R2

Dear Dr. Zhang,

We’re pleased to inform you that your manuscript has been judged scientifically suitable for publication and will be formally accepted for publication once it meets all outstanding technical requirements.

Kind regards,

Anas Bilal, Ph.D.

Academic Editor

PLOS ONE

Reviewers' comments:

Reviewer's Responses to Questions

**Comments to the Author**

1. If the authors have adequately addressed your comments raised in a previous round of review and you feel that this manuscript is now acceptable for publication, you may indicate that here to bypass the “Comments to the Author” section, enter your conflict of interest statement in the “Confidential to Editor” section, and submit your "Accept" recommendation.

Reviewer #1: (No Response)

Reviewer #3: (No Response)

2. Is the manuscript technically sound, and do the data support the conclusions?

Reviewer #1: Partly

Reviewer #3: Yes

3. Has the statistical analysis been performed appropriately and rigorously? 

Reviewer #1: Yes

Reviewer #3: Yes

4. Have the authors made all data underlying the findings in their manuscript fully available?

Reviewer #1: Yes

Reviewer #3: Yes

5. Is the manuscript presented in an intelligible fashion and written in standard English?

Reviewer #1: Yes

Reviewer #3: Yes

6. Review Comments to the Author

Reviewer #1: (No Response)

Reviewer #3: Please read the manuscript carefully and improve the grammar and language mistakes. It is suggested please improve the introduction and conclusion of the study.

7. PLOS authors have the option to publish the peer review history of their article (what does this mean?). If published, this will include your full peer review and any attached files.

Reviewer #1: **Yes: **Wenbo Zhu

Reviewer #3: No

---

## [Editor Report · Acceptance letter]

13 Oct 2023

PONE-D-23-16198R2 

A construction method of reverberation suppression filter using an end-to-end network 

Dear Dr. Zhang:

I'm pleased to inform you that your manuscript has been deemed suitable for publication in PLOS ONE. Congratulations! Your manuscript is now with our production department. 

Kind regards, 

on behalf of

Dr. Anas Bilal 

Academic Editor

PLOS ONE